# Visual statistical learning and integration of perceptual priors are intact in attention deficit hyperactivity disorder

**Katie L. Richards[1,2ᵒ], Povilas Karvelis[3ᵒ], Stephen M. Lawrie[1,4], Peggy Seriès[3]***

**1** Department of Psychiatry, Royal Edinburgh Hospital, University of Edinburgh, Edinburgh, United Kingdom,
**2** King's College London, Institute of Psychiatry, Psychology and Neuroscience, London, United Kingdom,
**3** Institute for Adaptive and Neural Computation, University of Edinburgh, Edinburgh, United Kingdom,
**4** Patrick Wild Centre, University of Edinburgh, Edinburgh, United Kingdom

ᵒ These authors contributed equally to this work.
* pseries@inf.ed.ac.uk

**Data Availability Statement:** The data are held in a public repository at https://osf.io/84zkb/.

**Funding:** KR is supported by the Health Foundation Scaling Up Award. PK was funded by Engineering

## Abstract

### Background

Deficits in visual statistical learning and predictive processing could in principle explain the key characteristics of inattention and distractibility in attention deficit hyperactivity disorder (ADHD). Specifically, from a Bayesian perspective, ADHD may be associated with flatter likelihoods (increased sensory processing noise), and/or difficulties in generating or using predictions. To our knowledge, such hypotheses have never been directly tested.

### Methods

We here test these hypotheses by evaluating whether adults diagnosed with ADHD ($n = 17$) differed from a control group ($n = 30$) in implicitly learning and using low-level perceptual priors to guide sensory processing. We used a visual statistical learning task in which participants had to estimate the direction of a cloud of coherently moving dots. Unbeknown to the participants, two of the directions were more frequently presented than the others, creating an implicit bias (prior) towards those directions. This task had previously revealed differences in other neurodevelopmental disorders, such as autistic spectrum disorder and schizophrenia.

### Results

We found that both groups acquired the prior expectation for the most frequent directions and that these expectations substantially influenced task performance. Overall, there were no group differences in how much the priors influenced performance. However, subtle group differences were found in the influence of the prior over time.

### Conclusion

Our findings suggest that the symptoms of inattention and hyperactivity in ADHD do not stem from broad difficulties in developing and/or using low-level perceptual priors.

and Physical Sciences Research Council. This study was supported by funding from the Patrick Wild Centre. SML reports receiving, in the past three years, personal fees from Janssen, Otsuka and Sunovion, and research funding from Janssen and Lundbeck. The other authors have no conflict of interest to declare. The funders had no role in study design, data collection and analysis, decision to publish, or preparation of the manuscript.

**Competing interests:** The authors have declared that no competing interests exist. As stated above, SML reports receiving, in the past three years, personal fees from Janssen, Otsuka and Sunovion, and research funding from Janssen and Lundbeck. This does not alter our adherence to PLOS ONE policies on sharing data and materials.

# Introduction

Attention deficit hyperactivity disorder (ADHD) is a common neurodevelopmental disorder characterized by age-inappropriate levels of inattention, hyperactivity, and/or impulsivity that substantially impact psychosocial functioning [1, 2]. The symptoms of ADHD have been hypothesized to stem from deficits in statistical learning and predictive 'top-down' processing [3]. Specifically, it has been proposed that disruptions in the development of frontostriatal and frontocerebellar neural loops result in difficulties in using temporal and contextual structure to guide cognition and behavior. This hypothesis of ADHD is in keeping with recent Bayesian predictive coding theories of neuropsychiatric disorders [4, 5].

Bayesian theories assume that cognition, from low-level sensory processing all the way through to higher-level beliefs, are governed by inferential processes [6–9]. In this view, perception is an active process, where percepts are generated by integrating noisy incoming sensory signals (*likelihood distribution*) with implicit beliefs or expectations about the state of the world (*prior distribution*). Bayes' rule is used to combine each source of information in a probabilistically 'optimal' manner, i.e. the most reliable (precise) source having the greatest influence upon perception. The prior acts as a summary of past experiences used to predict the most likely cause of sensation from noisy and ambiguous sensory data [10]. Errors originating from the comparison between predictions and incoming signals are used to update priors in order to minimize errors in future predictions [9]. The Prior probability distributions can be excessively precise or imprecise and failures in this precision (relative to that of the incoming signals) are thought to play an important role in the development of neuropsychiatric disorders [5].

There are numerous ways in which ADHD could be traced to differences in Bayesian predictive coding mechanisms. First, the failures of behavioral regulation in ADHD could be attributed to disruptions in the formation and/or use of priors, resulting in ascribing excessive precision to incoming information [3]. Specifically, characteristic symptoms, such as being easily distracted by external stimuli and difficulties maintaining prolonged attention on a task, could be due to excessive precision and therefore attention towards incoming sensory signals. Indeed, participants with ADHD exhibit diminished 'top-down' neural responses to expected stimuli, as well as enhanced early responses to sensory information and unexpected stimuli [11–14]. ADHD is also associated with a range of sensory modulation issues, including greater difficulties in using prior expectations to suppress unwanted saccades and reduce micro-saccade and blink rate around the onset of an anticipated stimulus [15–18]. Attenuated sensory priors and modulation issues could lead to a barrage of equally pertinent and intrusive sensations that cannot be habituated, resulting in distractibility and impulsive/hyperactive response patterns. Symptoms of inattention and over-activity have been shown to increase linearly with measures of atypical sensation in ADHD and the general population [19–21].

While reward learning deficits have been extensively studied in ADHD and are thought to arise from dopaminergic dysfunction [22–27], implicit learning has received very little attention in ADHD. Implicit learning is thought to play a crucial role in the formation of priors enabling a flexible and efficient interaction with the environment over short timeframes [28]. Investigations of implicit learning in ADHD are, however, mixed, with some studies finding a difference [29–32], whereas others do not [33–36]. Consistent evidence shows differences in frontostriatal and frontocerebellar circuitry in ADHD [37–39], areas implicated in implicit learning [40–42], lending support to the hypothesis that disruptions in implicit learning and consequently prior formation may account for ADHD symptomatology.

Second, elevated intra-individual variability has been outlined as a hallmark of ADHD and is evident in behavioral symptoms such as completing tasks in a muddled way [43]. ADHD is associated with notable increases in variability across cognitive domains, including perception

[44–47]. Such findings are suggestive of noisier and less precise distributions at the likelihood and/or prior level. However, it is unclear whether this variability originates from lower-level sensorimotor areas, higher-level cognitive regions, or both. Finally, the key symptom of ADHD, namely inattention, has been associated with a reduced gain in prediction error signals [48, 49]. Electrophysiological studies demonstrate reduced prediction error-related neural activity in ADHD, particularly error positivity, which is thought to represent an evaluation of prediction error [14].

Fine-grained computational models of Bayesian inferential processes are needed to tease apart these predictive coding mechanisms in ADHD. To test a Bayesian hypothesis of ADHD, we therefore used a visual statistical learning task, where participants estimate the direction of a cloud of coherently moving dots under varying levels of sensory uncertainty [50]. Unbeknown to participants, two of the directions are more frequently presented than the others, implicitly creating an expectation (prior) towards those directions. Previously, Chalk et al. [50] found that participants from the general population rapidly developed priors for the most frequent directions, and that these priors strongly influenced visual perception (i.e. perception was biased towards the most frequent direction). The performance of the participants was well described by a Bayesian model of sensory processing. These findings have since been replicated in a larger sample, in which higher autistic traits were associated with a weaker influence of the perceptual priors, due to a more precise representation of the sensory input [51].

Based upon the documented differences in ADHD we proposed the following hypotheses: individuals with ADHD may have difficulties in developing stable perceptual priors, perceptual priors may be noisier and/or their influence in guiding perceptual judgments may be weaker, resulting in a greater reliance upon incoming sensory information (the likelihood). Alternatively, or possibly additionally the representation of the sensory inputs might be noisier (sensory likelihoods would be less precise).

## Methods and materials

### Participants

Fifty participants (20 ADHD; 30 CTR) aged 18–65 years old were recruited from advertisements in primary care practices and educational settings. A consultant psychiatrist working within a specialist service for adults with ADHD also referred individuals to the study. Participants were included if they had normal or corrected-to-normal vision, were able to provide fully informed consent, and had an IQ > 70 (as measured by the Wechsler Abbreviated Scale of Intelligence; [52]). Diagnoses were verified using the Diagnostic Interview for ADHD in adults (DIVA; [53]). Sixteen of the ADHD participants presented with combined subtype and four with the predominantly inattentive subtype. Nine of the ADHD participants were taking stimulant medication and five were taking anti-depressants. Participants abstained from taking their stimulant medication on the day of testing. Participants with any neurological disorder, bipolar disorder, autism spectrum disorder, or psychotic disorders were excluded.

All participants were interviewed using the Structured Clinical Interview for DSM-IV (SCID-I; [54]) to determine inclusion/exclusion criteria, and completed the Adult ADHD Self-Report Scale v1.1 (ASRS; [55]) and Autism-Spectrum Quotient (AQ; [56]). The characteristics of the included participants are summarized in Table 1. The participant groups did not significantly differ in age, gender, or IQ. The ADHD group reported significantly higher autistic traits and ADHD symptoms, and substantially poorer functioning. The study received ethical approval from the South East Scotland Research Ethics Committee 01 and NHS Lothian Research & Development. Participants provided fully informed written consent and were financially compensated for their time and travel.

**Table 1. Participant characteristics (standard deviation in parentheses).**

|  | ADHD (*n* = 17) | CTR (*n* = 30) | Statistic | *p* |
|---|---|---|---|---|
| Age (years) | 34.12 (11.12) | 34.52 (11.12) | $Z = 0.13$ | .90 |
| Gender (M:F) | 8:9 | 19:11 | $\chi^2 = 1.18$ | .27 |
| Full-scale IQ | 122.53 (7.31) | 118.38 (7.21) | $Z = -1.52$ | .13 |
| Performance IQ | 120.13 (7.73) | 118.35 (9.55) | $Z = -0.48$ | .63 |
| Verbal IQ | 119.60 (6.31) | 114.65 (9.13) | $Z = -1.69$ | .09 |
| GAF | 66.71 (11.00) | 74.79 (10.70) | $Z = 2.40$ | .016 |
| AQ | 20.59 (6.30) | 13.37 (8.86) | $Z = -2.81$ | .005 |
| ASRS | 54.94 (9.36) | 29.79 (10.49) | $Z = -4.80$ | < .001 |

*Note.* ADHD = Attention Deficit Hyperactivity Disorder; CTR = Control; GAF = Global Assessment of Functioning; AQ = Autism-Spectrum Quotient; ASRS = Adult ADHD Self-Report Scale. Wilcoxon rank-sum test was used for all comparisons, except for gender balance comparison which used Chi square test.

## Apparatus, stimuli, & procedure

The setup for this study was similar to Chalk et al. [50] and is therefore only briefly described here. The stimuli were displayed on a Dell P790 monitor running at 1024 x 768 at 100Hz using MATLAB's Psychophysics Toolbox [57]. The visual stimuli consisted of a cloud of dots moving coherently (100%) within a circular annulus with a white fixation point in the center and a red bar extending out from this fixation-point (see Fig 1A). The visibility of the dots was altered throughout the task by presenting four randomly interleaved contrast levels: zero contrast (no stimulus) (167 trials), two low-contrast levels (90 trials at 2/1 staircase; 243 trials at 4/1 staircase), and one high-contrast level (67 trials). The contrast on high-contrast trials was 1.76 cd/m$^2$ above a 5.18 cd/m$^2$ background. The cloud of dots moved at 0˚, ±16˚, ±32˚, ±48˚, and ±64˚ with respect to a central reference angle. This central reference angle was randomized for each participant. Across all the low or high-contrast trials, the dots moved at ±32˚ for

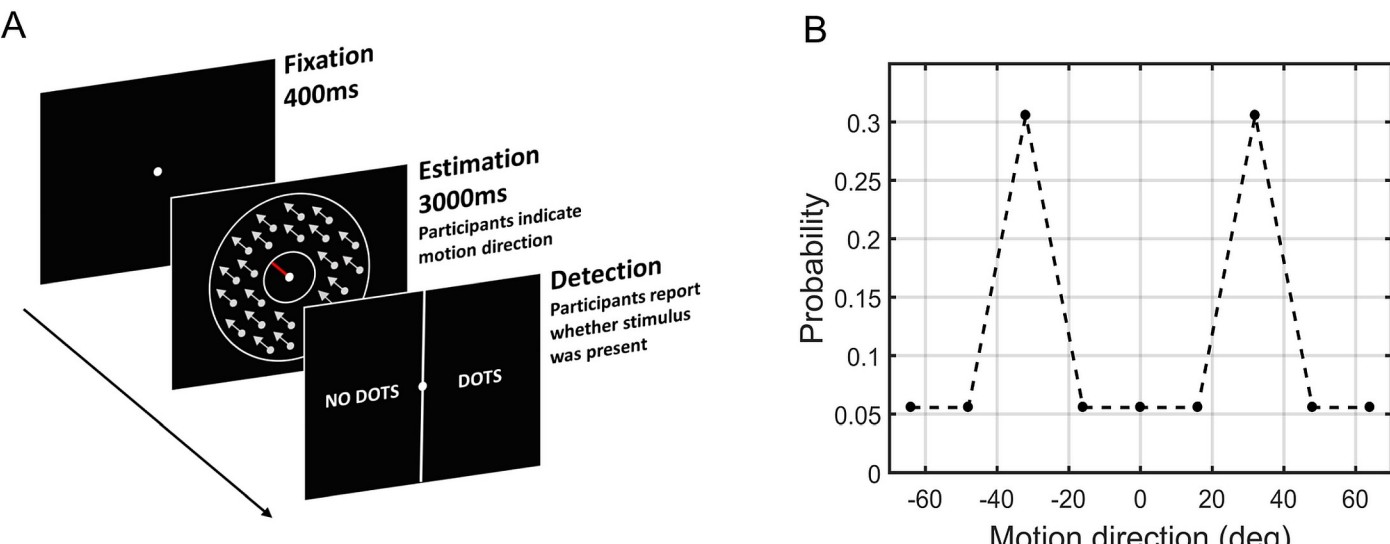

**Fig 1. The motion detection task.** (A) On each trial, participants were presented with a fixation point followed by a cloud of moving dots and a response bar (red bar). Participants were instructed to align the red bar to the direction the dots were moving in. The screen was cleared either when participants made an estimation or when 3000 ms had elapsed. Lastly, a new screen presented participants with a two-alternative forced choice task (2-AFC) between 'NO DOTS' or 'DOTS'. (B) Probability distribution of the motion directions. Unbeknownst to participants, the dots moved at ±32˚ more often than all the other directions.

58% of the trials, in the other predetermined directions (0˚; ±16˚; ±48˚; ±64˚) for 36% of the trials, and in completely random directions for 6% of the trials. The increased number of trials at ±32˚ created a bimodal probability distribution (Fig 1B), and thus a prior expectation that the dots would move at ±32˚. Participants were not told that stimuli would be presented more frequently at some directions than others.

Each trial was composed of two tasks, an *estimation task*, where participants indicated the direction of stimuli motion and a *detection task*, where participants reported whether they perceived any stimulus (Fig 1A). Participants received block-feedback every 20 trials on the accuracy of their estimation performance and immediate feedback for detection performance. The task was completed in a darkened room at ~100cm viewing distance. Participants completed 567 trials of the task with breaks every 170 trials (taking ~45 minutes to complete).

## Data analysis

### Behavioral data analysis

Performance on high-contrast trials was used as a benchmark to ensure adequate performance in the task. >70% detection and <30˚ estimation root mean square error (RMSE) were the inclusion criteria. Two ADHD participants did not meet these criteria; one more ADHD participant was excluded due to poor detection performance (<50%) on the low-contrast trials (S1 Fig in S1 File).

The 2/1 and 4/1 staircases converged to stable luminance levels after approximately 100 trials for both participant groups (S2 Fig in S1 File). There was no difference in the average luminance level achieved by the 2/1 and 4/1 staircases, and the data was combined across the staircases.

Estimation performance measures on low-contrast trials (2/1 and 4/1 staircases) were computed only from trials where an estimation response was made within the given time (3000 ms) and participants reported seeing dots. To compute estimation biases, variability and lapses, the estimation responses were fitted to a mixed circular normal distribution (von Mises and uniform distribution).

$$(1 - \alpha) \cdot V(\mu, \sigma) + {}^{\alpha}/_{2}\pi \qquad (1)$$

where $V(\mu, \sigma)$ is the von Mises circular normal distribution with mean $\mu$ and width $\sigma$. The estimation bias is calculated as the difference between $\mu$ and the true motion direction, while the estimation variability corresponds $\sigma$. Parameter $\alpha$ corresponds to the proportion of lapse estimations.

On no-stimulus trials participants occasionally experienced hallucinations. To quantify acquired prior effects on these responses, we computed a probability ratio that captured how much the participants hallucinated stimulus was moving within 16˚ of ±32˚ than at all other directions:

$$p_{ratio} = p(\theta_{est} = \pm32(\pm16)^{\circ}) \cdot N_{bins} \qquad (2)$$

where $N_{bins}$ = 11, is the number of bins across the whole response range. This probability ratio would be equal to 1 if participants were equally likely to estimate within 16˚ of ±32˚ as they were to estimate within 16˚ of the other bins.

A 2 (between-subject factor: ADHD, CTR) x 5 (within-subject factor: 0˚, ±16˚, ±32˚, ±48˚, and ±64˚) mixed ANOVA was used to determine the impact of the acquired prior on the estimation bias, variability, reaction time and detection performance across the groups. Post-hoc t-tests used Bonferroni-correction. The tests were conducted in SPSS version 25. Bayes factors (BF_{01}) were used to evaluate the strength of the evidence for the null hypothesis using the

Bayesian statistical software package JASP version 0.10. A Bayes factor between 1–3 indicates weak evidence, 3–10 indicates moderate evidence, and > 10 indicates strong evidence [58]. The analysis was re-ran for bias, variability, and hallucinations while controlling for AQ scores, as these measures were previously found to correlate with AQ [51]. Moreover, AQ and ASRS scores positively correlated across the groups ($p = 0.012$). The measures were positively correlated within controls ($p = 0.041$), while within the ADHD group there was a trend towards a negative correlation that did not reach significance ($p = 0.097$). AQ scores were z-transformed [59].

## Modelling

To control for the possibility of different mechanisms underlying the performance of each individual, we fitted a range of models to our data. The first class of models was Bayesian: on every trial, the incoming sensory information is combined with a learned prior, with the mean of the resulting posterior distribution corresponding to the percept. We tested four variants of the Bayesian models (detailed below). The second class of models assumed that task performance could be explained by response strategies that do not involve Bayesian integration [60]: on any given trial participants responded by relying on either the prior or the likelihood alone. The resulting response distribution is effectively a sum of the prior and the likelihood (hence the class name 'ADD'). We considered four variations of the 'ADD' model (see S1 File). Below we present only the Bayesian models as they provided a better explanation to the data. Model comparison and parameter estimation methods are in the S1 File.

## Bayesian models

Following the Bayesian framework, we assumed that participants combined sensory information (likelihood) with their expectations about the motion direction (prior) on every trial (Fig 2). The sensory likelihood of the observed motion direction ($\theta_s$) was parameterized as a von Mises circular normal distribution with variance $\sigma_s$:

$$p_{\text{likelihood}}(\theta_s|\theta) = V(\theta, \sigma_s) \tag{3}$$

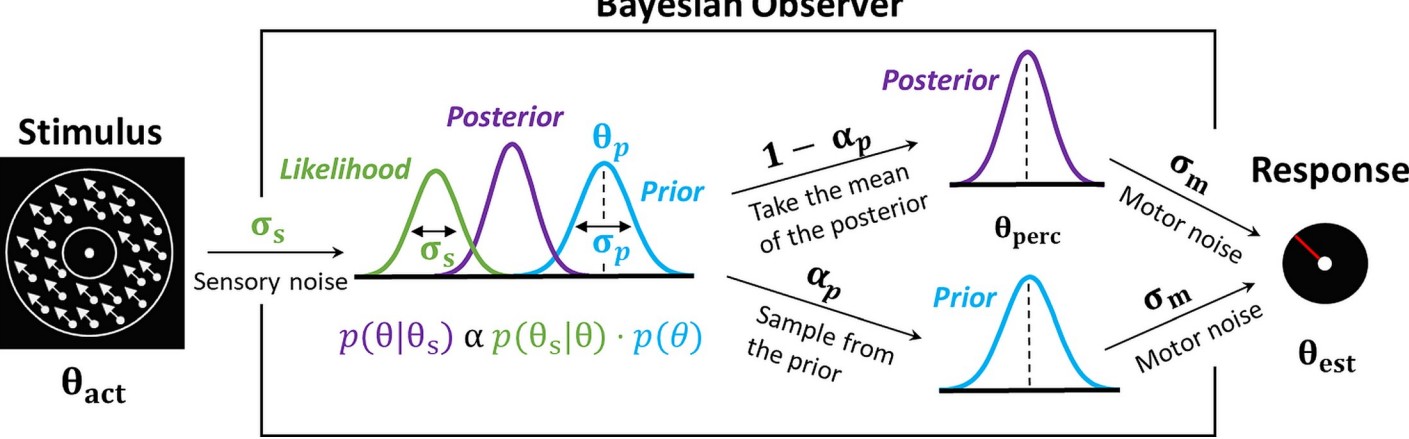

**Fig 2. Bayesian model of estimation response for a single trial for the best fitting model (Bayes_P).** The actual motion direction ($\theta$act) is corrupted by sensory uncertainty ($\sigma$s), and then combined with prior expectations (mean $\theta$p and uncertainty $\sigma$p) to form a posterior distribution. The mean of the posterior distribution then corresponds to the perceived motion direction ($\theta$perc). However, on a fraction of trials, determined by the prior-based lapses ($\alpha$p), the perceived motion direction is sampled directly from the prior. Finally, in both cases, the response ($\theta$est) is made by perturbing $\theta$perc with motor noise ($\sigma$m). This results in 4 free model parameters: $\sigma$s, $\sigma$p, $\theta$p and $\alpha$p. The motor noise ($\sigma$m) is estimated from high contrast trials and is used as a fixed parameter during the model fitting.

The mean of this distribution depended on the actual presented motion direction ($\theta_{act}$), and to account for trial-to-trial variability it was drawn from another von Mises distribution V ($\theta_{act}$, $\sigma_s$) centered on $\theta_{act}$ with variance $\sigma_s$.

We then hypothesized that participants acquired priors ($p_{prior}$ ($\theta$)) that approximated the bimodal distribution of the stimulus statistics. These priors were parameterized as the sum of two von Mises distributions, centered on motion directions $\theta_p$ and $-\theta_p$, each with variance $\sigma_p$:

$$p_{prior}(\theta) = \frac{1}{2} [V(-\theta_p, \sigma_p) + V(\theta_p, \sigma_p)] \tag{4}$$

Combining the prior and the likelihood gives us the posterior probability that the stimulus is moving in a direction $\theta$:

$$p_{posterior}(\theta|\theta_s) \propto p_{likelihood}(\theta_s|\theta) \cdot p_{prior}(\theta) \tag{5}$$

The perceived direction, $\theta_{perc}$, was taken to be the mean of the posterior distribution.

Finally, we accounted for motor noise and lapse estimations (random responses), such that:

$$p(\theta_{est}|\theta_{perc}) = (1 - \alpha_p) \cdot V(\theta_{perc}, \sigma_m) + \alpha_p \cdot [p_{prior}(\theta) * V(0, \sigma_m)] \tag{6}$$

where the asterisk (*) denotes convolution, $\sigma_m$ is the motor noise and $\alpha_p$ is the probability of prior-based lapse estimations (i.e. lapse estimations that follow the participants' acquired expectations–$p_{prior}(\theta)$). We called this model 'BAYES_P' for Bayes with Prior-based lapses (Fig 2).

We also tested a simpler variant of this model which assumed that the lapse estimations (Eq (5)) were not made based on the acquired prior but instead were completely random (model 'BAYES'). Furthermore, to account for the possibility of adaptations in the sensory likelihood itself (e.g., [61]), we tested two other variants of this model: 'BAYES_var' where the sensory precision varied with each stimulus direction and 'BAYES_varmin' where sensory precision was allowed to be different for ±32˚ but was the same for all other directions. BAYES_P and BAYES had a total of 4 free parameters, while BAYES_varmin and BAYES_var had 5 and 8, respectively.

## Results

### Behavioral data analysis

**Performance on low-contrast trials.** *Attractive bias.* First, we investigated participants' performance in the estimation of the direction of the moving stimuli, and more particularly the level of attractive bias towards ±32˚ at each of the predetermined motion directions (0˚, ±16˚, ±32˚, ±48˚, ±64˚). Fig 3A displays the average estimation bias plotted against the presented motion direction for each group. Overall, there was a significant effect of motion direction ($F(2.58, 115.93) = 10.15$, $p < .001$, $\eta_p^2 = 0.184$, Greenhouse-Geisser correction $\varepsilon = 0.644$), but no differences between the groups ($F(1, 45) = 0.17$, $p = 0.681$, $\eta_p^2 = 0.004$; $BF_{01} = 4.69$); and no group*angle interaction effect ($F(2.58, 115.93) = 1.86$, $p = .148$, $\eta_p^2 = 0.040$). Furthermore, controlling for AQ scores showed no differences in groups ($F(1, 33) = 0.32$, $p = 0.578$, $\eta_p^2 = 0.009$; $BF_{01} = 4.58$) and there was no correlation between mean bias and ASRS ($\tau_b = -0.16$, $p = .173$; $BF_{01} = 1.83$). Pairwise comparisons revealed that there was an attractive bias towards ±32˚ at ±64˚ (mean difference ($M_{diff}$) = 10.12, $p = .001$), at ±48˚ ($M_{diff} = 3.63$, $p = 0.015$) and at ±16˚ ($M_{diff} = -2.72$, $p = 0.036$).

**Variability.** We also evaluated whether the perceptual prior influenced the variability of estimation responses at each of the predetermined motion directions (Fig 3B). We found a

significant main effect of motion direction ($F(2.87, 128.99) = 5.70$, $p = .001$, $\eta_p^2 = 0.112$, Greenhouse-Geisser correction $\varepsilon = 0.717$), but no differences between the groups ($F(1, 45) = 0.01$, $p = .750$, $\eta_p^2 < 0.001$; $BF_{01} = 3.62$); and no group*angle interaction effect ($F(2.87, 128.99) = 0.86$, $p = .461$, $\eta_p^2 = 0.019$). Furthermore, controlling for AQ scores showed no differences in groups ($F(1, 33) = 0.02$, $p = 0.887$, $\eta_p^2 = 0.001$; $BF_{01} = 3.25$) and there was no correlation between mean variability and ASRS ($\tau_b = 0.13$, $p = .288$; $BF_{01} = 2.62$). Pairwise comparisons revealed that the effects were driven by the variability at ±32° being lower than at 0° ($M_{diff} = 4.77$, $p = .008$), at ±16° ($M_{diff} = 2.84$, $p = .007$) and at ±64° ($M_{diff} = 3.32$, $p = .041$).

**Reaction time.** Next, we examined whether the reaction time varied across the predetermined motion directions (Fig 3D). There was a significant main effect of motion direction on reaction time ($F(2.71, 121.77) = 9.45$, $p < .001$ $\eta_p^2 = 0.174$, Greenhouse-Geisser correction $\varepsilon = 0.677$). This was driven by decreased reaction times at the most frequent directions, reaction time at ±32° was significantly shorter than at all other directions (0°, $M_{diff} = 0.09$, $p = .015$; ±16°, $M_{diff} = 0.05$, $p < .033$; ±48°, $M_{diff} = 0.06$, $p < .014$; ±64°, $M_{diff} = 0.14$, $p < .001$). There was no significant main effect of group on reaction time ($F(1, 45) = 3.40$, $p = .072$, $\eta_p^2 = 0.070$), and there was no interaction between group and motion direction ($F(2.71, 121.77) = 1.28$, $p = .284$, $\eta_p^2 = 0.028$). There was no correlation between mean reaction time and ASRS ($\tau_b = -0.19$, $p = .102$; $BF_{01} = 1.22$).

*Detection.* Finally, we analyzed whether the acquired prior improved detection at the expected motion directions (Fig 3E). There was a significant main effect of motion direction

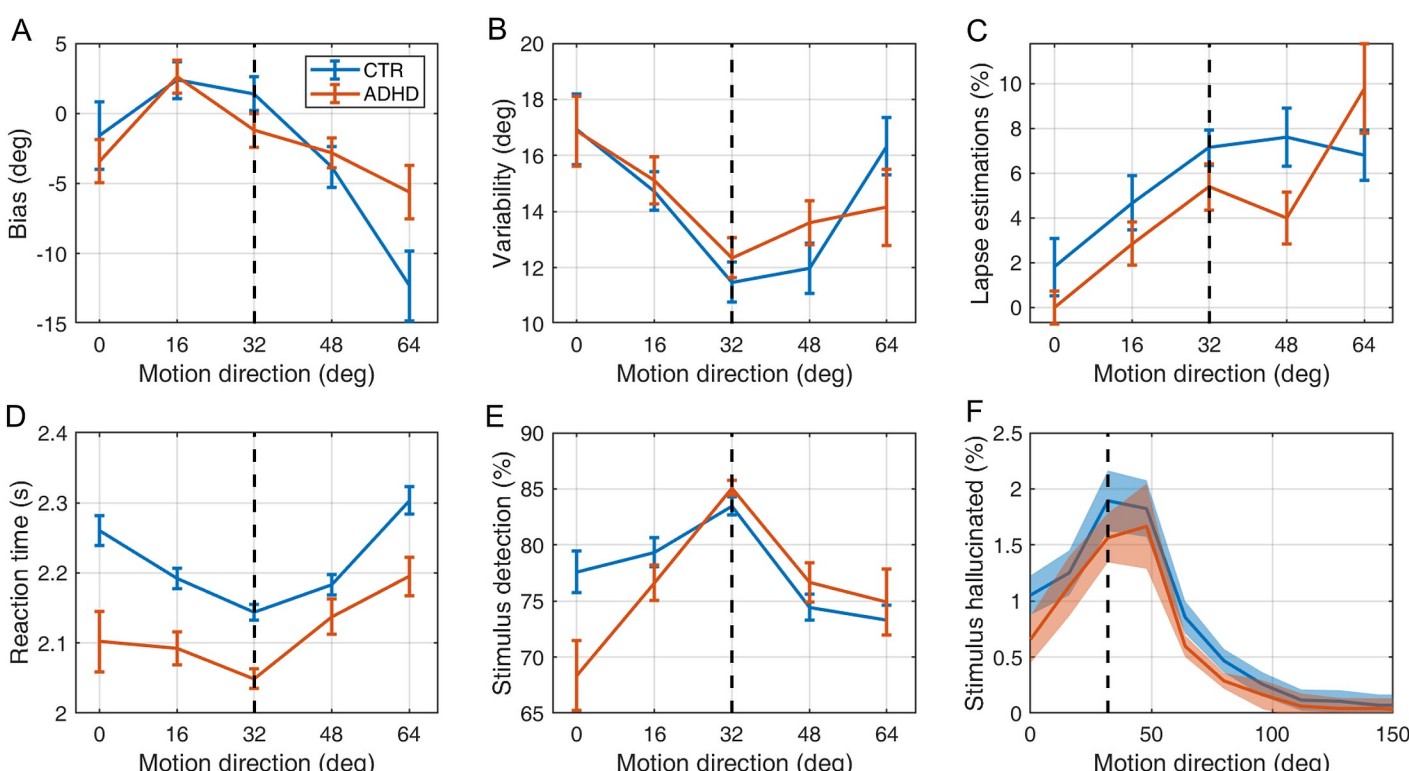

**Fig 3.** Performance on (A-E) low contrast trials and (F) no stimulus trials by CTR (blue lines) and ADHD participants (orange lines). (A) Mean estimation bias (B) estimation variability (C) lapse estimations determined using (Eq 1), (D) reaction times during the estimation task, (E) the fraction of trials in which the stimulus was detected, (F) the fraction of no stimulus trials in which the stimulus was hallucinated. The error bars and shaded areas represent within-subject standard error. The vertical dashed lines correspond to the most frequently presented motion directions (i.e. ±32°).

on detection performance ($F(2.34, 105.26) = 11.31$, $p < .001$, $\eta_p^2 = 0.201$, Greenhouse-Geisser correction $\varepsilon = 0.585$), with stimulus at ±32° being detected more frequently than at all other directions (0°, $M_{diff} = 11.23$, $p < .001$; ±16°, $M_{diff} = 6.30$, $p < .001$; ±48°, $M_{diff} = 8.87$, $p < .001$; ±64°, $M_{diff} = 10.58$, $p < .001$). The main effect for group is not presented as the contrast staircases guarantees that all participants have the same average detection rate, but there was a significant group*motion direction interaction: $F(2.34, 105.26) = 3.02$, $p = .045$, $\eta_p^2 = 0.063$, which was driven by controls having better detection at 0°($M_{diff} = 9.24$, $p = .019$).

Finally, we also examined the dynamics of prior learning (see S1 File). The effect of the prior became significant for both groups within 110 trials for estimation bias, detection, and reaction time (S3 Fig in S1 File). While group differences in the acquisition of the prior were largely non-significant, ADHD participants did demonstrate significantly stronger prior effects on detection rate towards the end of the task and showed less estimation bias than controls in the middle of the task (between trials 220 to 330).

**Perceived motion in absence of visual stimuli ('hallucinations').**   In a number of trials, in absence of a visual stimulus, both groups reported perceiving visual motion. We found that the median value of '$p_{ratio}$' was significantly greater than 1 for both participants groups ($Mdn$ ($p_{ratio}$) = 2.53, $p = .001$ and $Mdn(p_{ratio}) = 3.00$, $p < .001$, respectively; two-tailed signed-rank), indicating that both groups' hallucinations corresponded significantly more often to perceived motion around the most frequent motion directions as opposed to all other directions (Fig 3F). Bayesian statistical analysis provided evidence for no group differences ($BF_{01} = 3.29$). The groups did not differ in the number of total hallucinations experienced in the task ($Z = 0.12$, $p = .903$, two-tailed rank-sum; $BF_{01} = 3.01$). Finally, the correlation between the number of hallucinations and ASRS was not significant ($\tau_b = 0.14$, $p = .234$; $BF_{01} = 2.23$).

**Modelling results.**   We evaluated our models using Bayesian Information Criterion (BIC). We used two different methods for model comparison: fixed-effects approach, which sums BIC across individuals, and random effects Bayesian model comparison, which considers the distribution of BIC values across individuals. Both methods suggested BAYES_P model (Fig 2) to be superior (Fig 4). While parameter recovery analysis showed high recoverability of model parameters (see S1 File), visual inspection suggested that the BAYES_P fit to the data was not

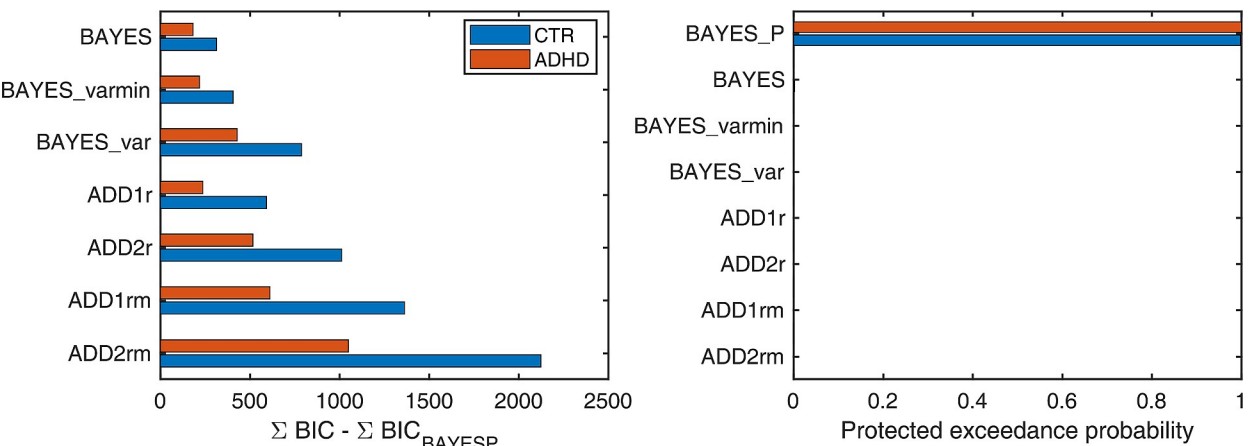

**Fig 4. Model comparison and selection.** (A) Fixed effects model selection using Bayesian Information Criterion (BIC). X-axis measures the relative difference between BIC of each model (as indicated on Y-axis) and BIC of BAYES_P (winning model) summed across participants. Smaller BIC indicate a better model. For both ADHD and control participants BAYES_P provided the best model evidence. (B) Random effect Bayesian model selection. Higher protected exceedance probability indicates a model having a higher likelihood of being more frequent among the subjects. For both ADHD and controls BAYES_P was the most likely model.

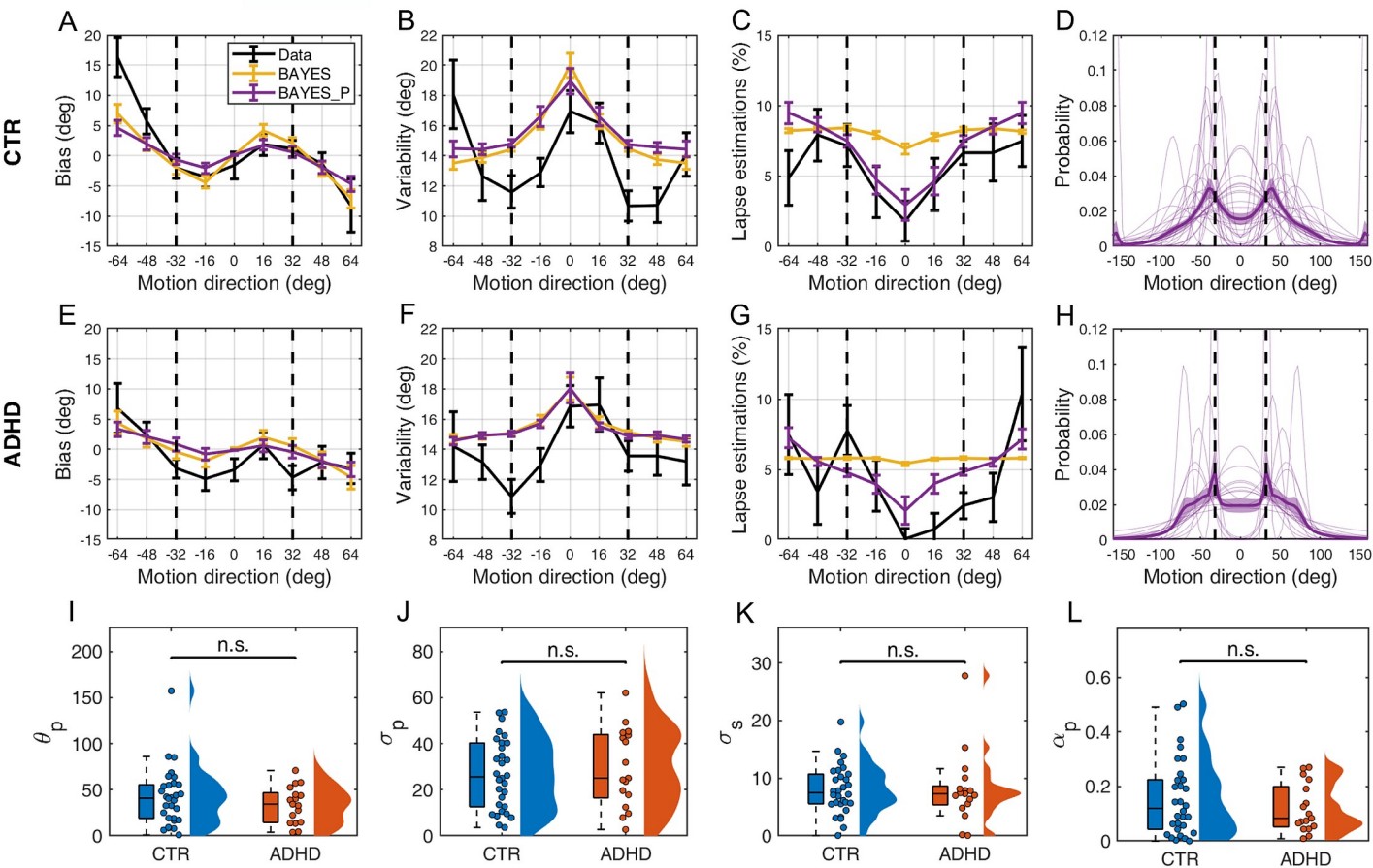

**Fig 5. Model fits and parameter estimates.** (A-H) Model fits for the best fitting model BAYES_P (purple) and the second-best model BAYES (yellow), to the behavioral data (black). (A-D) CTR and (E-H) ADHD participants. (A, E) Estimation bias, (B, F) estimation variability, (C, G) estimation lapse rate, (D, H) prior expectations of each individual (thin purple lines) and group average (thick purple line) as estimated via BAYES_P model. The vertical dashed lines correspond to the most frequently presented motion directions (i.e. ±32˚). The error bars and shaded areas represent within-subject standard error. (I-L) Comparison of BAYES_P model parameter estimates of CTR and ADHD participants; jittered dots denote individual participants; colored areas represent density of the data points. (I) θp–the mean of acquired prior (W = 220, p = .449, BF01 = 2.67), (J) σp–the uncertainty in the acquired prior (W = 282, p = .561; BF01 = 2.70), (K) σs–the uncertainty of sensory likelihood (W = 244, p = .818, BF01 = 3.01), (L) αp–prior-based lapse rate (W = 231, p = .606, BF01 = 3.30). n.s. = non-significant.

perfect (Fig 5A–5C and 5E–5G), warranting some caution in the interpretation of modelling results. Parameter recovery for the BAYES_P model is presented in the S1 File.

Finally, we compared the groups on BAYES_P parameters (Fig 5I–5L). Consistent with the behavioral data results, none of the parameters were different between the groups: the mode of the prior (W = 220, $p$ = .449, $BF_{01}$ = 2.67), the precision of the prior (W = 282, $p$ = .561; $BF_{01}$ = 2.70), the precision of the sensory likelihood (W = 244, $p$ = .818, $BF_{01}$ = 3.01) and the prior-based lapse estimations (W = 231, $p$ = .606, $BF_{01}$ = 3.30). Similarly, ASRS did not correlate with any of these model parameters: prior mean ($\tau_b$ = -0.21; $p$ = .076), prior uncertainty ($\tau_b$ = 0.09; $p$ = .478), sensory uncertainty ($\tau_b$ = -0.03; $p$ = .827), prior-based lapse rate ($\tau_b$ = 0.23; $p$ = .056)

## Discussion

This study used a visual statistical learning task to establish whether adults diagnosed with ADHD differed from a control group in rapidly learning and using low-level perceptual priors

to guide sensory processing. From a Bayesian perspective, we hypothesized that ADHD would be associated with difficulties in developing and/or using priors and therefore rely more on incoming sensory information in percept formation, or alternatively, that the representation of the sensory information might be noisier. Overall, we did not find evidence in support of any of these hypotheses. We found that both groups learned to expect the most frequent directions (the perceptual priors) and that these expectations strongly influenced task performance, replicating previous findings [50]. Both ADHD and control participants demonstrated faster reaction times, reduced variability, and better detection rates at the most frequent directions (±32˚), as well as an attractive estimation bias towards those directions. Moreover, in trials where no stimulus was actually present, both groups were more likely to report seeing dots moving at ±32˚ than at any other direction (hallucinations). There were no group difference and ADHD symptomatology did not influence any aspect of task performance. The performance of both groups was best described by a Bayesian model of sensory processing (similar to [50, 51, 62]). While the model did not provide an ideal fit warranting some caution in the interpretation of the results, it supported the behavioral data analysis showing no difference between groups in model parameters (prior mean, prior uncertainty, sensory likelihood uncertainty and prior-based lapse rate).

These findings are in keeping with evidence of intact statistical learning in decision-making, and sequential and spatial learning tasks in ADHD [33–36, 63]. Our results build upon previous work by using detailed computational models of implicit learning at an early stage of sensory processing in adults with ADHD. Statistical learning studies in adults diagnosed with ADHD are relatively rare (e.g. [34]) and most use implicit motor rather than perceptual learning tasks. The observed differences in learning reported in the literature also tend to be subtle or related to specific aspect of the task. For example, Barnes et al. [29] found reduced implicit sequence learning in ADHD relative to controls, but this difference was primarily driven by a reduced sensitivity to learning in the middle of the task but not at the start or end. In agreement with this, we also found subtle group differences in learning across time, specifically, that participants with ADHD showed slightly weaker prior estimation biases in the middle of the task and a stronger detection bias towards the end of the task.

The current findings are, however, at odds with studies showing that ADHD is associated with disruptions in neural systems that underlie implicit learning and predictive processing [14, 64, 65]. Similar to statistical learning paradigms, most neurophysiological and imaging studies have been conducted with samples of children participants rather than adults and focused on motor tasks or tasks requiring higher-level cognitive functions, such as inhibition [14, 38, 66]. The current study focused exclusively on low-level visual processing. It is still therefore plausible that ADHD could stem from difficulties in Bayesian predictive mechanisms at a higher-level of the cognitive hierarchy. Furthermore, differences at the neural level do not always result in observable differences at the behavioral level in ADHD (e.g. [12, 67]). It is also conceivable that a more complex prior distribution or a task that results in slower acquisition of the prior might allow differences in inferential processes to emerge.

Groups of individuals with ADHD are behaviorally, cognitively, and functionally heterogeneous [68, 69]. Although substantial efforts were made to recruit participants as broadly as possible from clinical and non-clinical settings, our sample was largely composed of participants with above average or superior intelligence that were either in full-time employment or education. Deficits in Bayesian inference could therefore exist in different subgroups of individuals with ADHD. Future studies, with larger, more heterogeneous samples are warranted to evaluate the degree to which the current findings can be generalized to the broader ADHD population. Another limitation of the current study is that many of the participants had or were currently taking stimulant medications. Although a washout period was used, it is not

feasible to eliminate the cumulative effects of stimulant medication on the brain [70, 71]. Our exploratory analysis of those participants that were currently taking stimulants and those that were not, however, did not suggest stimulant medication to have strong effects on our findings (see S1 File).

This study contributes to the growing body of evidence evaluating Bayesian hypotheses of neuropsychiatric disorders. To the best our knowledge, this is the first study to explicitly test differences related to Bayesian inference in ADHD. Our findings demonstrate that adults with ADHD develop and use low-level perceptual priors in a similar manner as controls during visual motion perception. Findings such as this, suggest that ADHD is not associated with a broad deficit in Bayesian inferential processes that extend all the way through the cognitive hierarchy from low-level sensory processing to higher-level functions. However, further testing is warranted in larger, more heterogeneous samples, and with more complex experimental tasks.

## Supporting information

**S1 File. Supplementary materials.**
(PDF)

## Acknowledgments

We sincerely thank Dr Prem Shah for assisting in recruiting participants and all the participants that contributed to this project.

## Author Contributions

**Conceptualization:** Stephen M. Lawrie, Peggy Seriès.

**Data curation:** Katie L. Richards.

**Formal analysis:** Povilas Karvelis.

**Funding acquisition:** Stephen M. Lawrie.

**Investigation:** Katie L. Richards.

**Methodology:** Katie L. Richards, Povilas Karvelis, Stephen M. Lawrie.

**Project administration:** Katie L. Richards, Stephen M. Lawrie, Peggy Seriès.

**Resources:** Stephen M. Lawrie.

**Software:** Povilas Karvelis, Peggy Seriès.

**Supervision:** Stephen M. Lawrie, Peggy Seriès.

**Visualization:** Katie L. Richards, Povilas Karvelis.

**Writing – original draft:** Katie L. Richards.

**Writing – review & editing:** Povilas Karvelis, Stephen M. Lawrie, Peggy Seriès.

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
