## [Decision Letter · Decision Letter 0]

29 Sep 2020

PONE-D-20-20668

Visual statistical learning and integration of perceptual priors are intact in Attention Deficit Hyperactivity Disorder

PLOS ONE

Dear Dr. Series,

Thank you for submitting your manuscript to PLOS ONE. After careful consideration, we feel that it has merit but does not fully meet PLOS ONE’s publication criteria as it currently stands. Therefore, we invite you to submit a revised version of the manuscript that addresses the points raised during the review process.

Both reviewers are quite positive about your manuscript. Nevertheless they require some minor conerns to be addressed before recommeding the manuscript fpr publication.

We look forward to receiving your revised manuscript.

Kind regards,

Thilo Kellermann, PhD

Academic Editor

PLOS ONE

Journal Requirements:

'KR is supported by the Health Foundation Scaling Up Award. PK was funded

by Engineering and Physical Sciences Research Council. This study was supported by funding from

the Patrick Wild Centre.'

'SML reports receiving, in the past three years, personal fees from Janssen, Otsuka and Sunovion, and research funding from Janssen and Lundbeck. The other authors have no conflict of interest to declare.'

5. Please include captions for your Supporting Information files at the end of your manuscript, and update any in-text citations to match accordingly. Please see our Supporting Information guidelines for more information: http://journals.plos.org/plosone/s/supporting-information

Reviewers' comments:

Reviewer's Responses to Questions

**Comments to the Author**

1. Is the manuscript technically sound, and do the data support the conclusions?

Reviewer #1: Yes

Reviewer #2: Yes

2. Has the statistical analysis been performed appropriately and rigorously? 

Reviewer #1: Yes

Reviewer #2: Yes

3. Have the authors made all data underlying the findings in their manuscript fully available?

Reviewer #1: Yes

Reviewer #2: Yes

4. Is the manuscript presented in an intelligible fashion and written in standard English?

Reviewer #1: Yes

Reviewer #2: Yes

5. Review Comments to the Author

Reviewer #1: Richards 2020

The researchers aimed to test whether there are perturbances in Bayesian inference in individuals with ADHD which might explain the symptoms they experience. During a dot-motion detection task, participants were required to indicate the angle of the movement of the dots, as well as whether the stimulus was present in the first place. Stimuli were mostly like to be moving towards 32 degrees with respect to a reference point, and less likely at 4 other locations either 16 or 32 degrees from the most likely reference point in either direction, creating 5 motion conditions. Behavioural analysis as well as a modelling analysis was used to explore differences between groups. The researchers found a bias towards reporting motion in the most likely direction. However, there were no group differences in these biases. Computational modelling further did not reveal any differences across groups. The paper is well written and the methods are sound. There are a few concerns I would like to raise, but I don’t think they will stand in the way of getting the paper published after some revisions.

Major:

1. The introduction could benefit from a bit more information on how supposedly reduced priors in perceptual inference map onto the symptoms of ADHD specifically, in particular in the presence of a null-finding, and the researchers discussing a similar hypothesis for autism.

2. The task uses a stair-case procedure to control for each subject when the dots are visible. I’m wondering whether this not eliminates a possible effect of sensory precision, and if not why not?

3. Given effects of dopamine on the usage of prior expectations in perceptual inference (see Cassidy et al., 2018 for example), I wonder to what extend medication history affects the results. I realise it is a small sample, but it would be good to do an exploratory analysis to see how effects differ between those with a history of stimulant use between those who do not.

Minor:

1. Why is there not a bias found towards the 32 angle in the 0degree condition. It is equally far from the 64 condition where the strongest effect is found, so it is surprising there is no effect in what otherwise is a very similar condition.

2. With regards to the task, could you clarify whether participants had to do the estimation and detection task simulataneously or sequantially? Do they require the same button response? How does this work exactly?

3. The results are controlled by AQ scores which makes sense, but is there a correlation in the first place?

Reviewer #2: This well-written paper depicts an important study that applies a visual statistical learning task and associated models developed by this research group (Chalk et al, 2010, Parvelis et al, 2018, Valton et al, 2019) to investigate potential impairments in Bayesian inference in ADHD. As the authors emphasize, this study is the first to probe directly Bayesian inference in ADHD and utilize a task and model capable of finding out if ADHD participants have less precise sensory representations or likelihoods or deficits in forming (or using) predictions or priors. This is a valuable endeavor as there is increasing evidence of alterations in Bayesian inference in several neuropsychiatric disorders.

This paper has very many additional strengths, both conceptual and methodological: it is one of the few investigations of implicit learning in ADHD, the writing is clear and compelling, the analyses and models are thoroughly presented and executed and thus convincing (i.e, individual data points are presented in several plots, there are 2 different methods for model comparison that lead to the same winning model, parameter recovery is amazing). Overall, I am very happy to see this powerful paradigm applied to ADHD and I believe this article is great and enthusiastically recommend it for publication. However, beforehand, I would like to see a few aspects developed and unpacked in more detail.

Specific comments.

1. Overall, I believe the introduction is nicely written and thorough and helps place this work within the broader literature of ADHD and computational psychiatry. However, the authors could also more explicitly allude to the larger literature of reward learning in ADHD and contrast with implicit learning, which can plausibly also be impaired in ADHD, but has been understudied. Something along the lines of: While impairments in reward learning and response have been extensively characterized in ADHD and are thought to stem from dopaminergic dysfunction (for instance, Frank et al, 2007, Silvetti et al, 2013, Kollins and Adcock, 2014, Ziegler et al, 2016, Chevrier et al, 2019), fewer studies have investigated implicit learning in ADHD. The ability to implicitly learn the statistics of a given environment in a short time frame allows for flexible and effective interaction with and function within that environment.

2. page 10. “The evidence of difficulties in implicit learning, crucial for the development of priors, is more mixed in ADHD, with some studies finding a difference [19-22], whereas others do not [23-26]. “ In the discussion, the authors do justice to the possibility that the effects of medication might account for the lack of difference between Controls and ADHD (even if ADHD participants did not take their medication the day of the experiment, the washout period is expected to be longer than that). Are there differences in the medication status of the participants from the studies which found differences in implicit learning in ADHD vs the ones which did not?

3.page 10. Citations [27-29] seem more suitable to be placed after “ADHD” vs where they are now: “There is also consistent evidence showing differences in frontostriatal and frontocerebellar circuitry in ADHD, areas implicated in implicit learning [27-29]. “ Other citations would support that these areas are also implicated in implicit learning. (i.e. Turke-Browne et al, 2009, Yang and Li, 2012, Leow et al, 2017)

4. Results related to Figure 5. The authors are testing several alternative models and the Bayes_P model convincingly captures the data best out of the models tested. The Bayes_P model seems to capture the data well and we can see this visually in Figure 5. I was wondering about two things. First, it seems that in Chalk et al, 2010 and in Karvelis et al, 2018 the Bayes model with a random lapse was able to capture the data very well, but in the Valton et al, 2019 and this paper the Bayes P with the prior-based lapses outperformed the Bayes model, even in Controls. What could account for this difference? One difference I can see is that the lapse parameters were estimated at lower values in Karvelis vs this paper, and thus accounted for a smaller part of the data? Second, while the model fits overlap with the data well, they do not seem quite as impressive as in Chalk or Karvelis. Is this fair to say? If so, what might account for these data sets being harder to fit? Relatedly, this might warrant a bit more caution in the interpretation of the model parameters.

5. Page 19 .Last sentence of results. The authors write “Similarly, ASRS did not correlate with any of these model parameters. “, but do not provide values.

6. Page 19. Discussion, end of the first paragraph: “there were no group difference in model parameters. “ It is worth spelling out again the model parameters: the mean of the acquired prior, the uncertainty of the prior or the uncertainty of the likelihood. Also, perhaps consider adding that while caution is warranted in the interpretation of individual model parameters, the goodness-of-fits of the Bayes_P model was comparably good across the ADHD and Control groups?

7. Page 21. The authors end with “further testing is warranted in larger, more heterogeneous samples, and with more complex experimental tasks.” In the Chalk paper, the authors write in the discussion: “Future work could investigate this using a more complicated distribution of presented stimuli or statistical learning paradigm that produces slower learning of stimulus expectations (Eckstein et al., 2004; Orba ´n, Fiser, Aslin, & Lengyel, 2008). “ I believe such a discussion point might benefit from a bit more emphasis here, as impairments in ADHD might be revealed in tasks with increased complexity of the prior distribution to be learnt and used, or with slower tasks. The authors’ finding of differences in the influence of the prior over time lends support to this possibility.

6. PLOS authors have the option to publish the peer review history of their article (what does this mean?). If published, this will include your full peer review and any attached files.

Reviewer #1: No

Reviewer #2: No

---

## [Editor Report · Decision Letter 1]

16 Nov 2020

Visual statistical learning and integration of perceptual priors are intact in Attention Deficit Hyperactivity Disorder

PONE-D-20-20668R1

Dear Dr. Series,

We’re pleased to inform you that your manuscript has been judged scientifically suitable for publication and will be formally accepted for publication once it meets all outstanding technical requirements.

Please check, if your required changes regarding the statements for "Financial Disclosure" and "Competing Interests" have been adopted correctly, because you mentioned these changes in your letter only (with no changes made in the respective form). I have sent a corresponding query to the staff of PLOS ONE and I hope that they will take care of this issue.

Kind regards,

Thilo Kellermann, PhD

Academic Editor

PLOS ONE
---

## [Editor Report · Acceptance letter]

9 Dec 2020

PONE-D-20-20668R1 

Visual statistical learning and integration of perceptual priors are intact in Attention Deficit Hyperactivity Disorder 

Dear Dr. Series:

I'm pleased to inform you that your manuscript has been deemed suitable for publication in PLOS ONE. Congratulations! Your manuscript is now with our production department. 

Kind regards, 

on behalf of

Dr. Thilo Kellermann 

Academic Editor

PLOS ONE